# Quantitative Sensing of Domoic Acid from Shellfish Using Biological Photonic Crystal Enhanced SERS Substrates

**DOI:** 10.3390/molecules27238364

**Published:** 2022-11-30

**Authors:** Subhavna Juneja, Boxin Zhang, Nabila Nujhat, Alan X. Wang

**Affiliations:** 1School of Electrical Engineering and Computer Science, Oregon State University, Corvallis, OR 97331, USA; 2Department of Electrical and Computer Engineering, Baylor University, Waco, TX 76798, USA

**Keywords:** marine biotoxin, biosensor, SERS, photonic crystals, chemometric analysis

## Abstract

Frequent monitoring of sea food, especially shellfish samples, for the presence of biotoxins serves not only as a valuable strategy to mitigate adulteration associated health risks, but could also be used to develop predictive models to understand algal explosion and toxin trends. Periodic toxin assessment is often restricted due to poor sensitivity, multifarious cleaning/extraction protocols and high operational costs of conventional detection methods. Through this work, a simplistic approach to quantitatively assess the presence of a representative marine neurotoxin, Domoic acid (DA), from spiked water and crab meat samples is presented. DA sensing was performed based on surface-enhanced Raman scattering (SERS) using silver nanoparticle enriched diatomaceous earth—a biological photonic crystal material in nature. Distinctive optical features of the quasi-ordered pore patterns in diatom skeleton with sporadic yet uniform functionalization of silver nanoparticles act as excellent SERS substrates with improved DA signals. Different concentrations of DA were tested on the substrates with the lowest detectable concentration being 1 ppm that falls well below the regulatory DA levels in seafood (>20 ppm). All the measurements were rapid and were performed within a measurement time of 1 min. Utilizing the measurement results, a standard calibration curve between SERS signal intensity and DA concentration was developed. The calibration curve was later utilized to predict the DA concentration from spiked Dungeness crab meat samples. SERS based quantitative assessment was further complemented with principal component analysis and partial least square regression studies. The tested methodology aims to bring forth a sensitive yet simple, economical and an extraction free routine to assess biotoxin presence in sea food samples onsite.

## 1. Introduction

Marine toxin from booming algae has been alarming in the recent decades as it poses significant threats to shellfish industry. Algae is unicellular plant-resembling phytoplankton that utilizes sunlight, carbon dioxide and water to photosynthesize organic nutrients, essential for maintaining a healthy marine ecosystem [1]. Anthropogenic eutrophication from activities such as untreated sewage, dissolved chemicals, fertilizer runs offs, etc., has however, triggered adverse algal occurrence, in frequency, duration and density [2]. Consequential outcomes of algal plague include altering current rates and pH of the water bodies, loss of marine life, increased production of toxin releasing algae species and, subsequently, contaminated food and water resources. The Pacific Northwest bounded by the coastal waters is no exception to the occurrence of harmful algal blooms (HABs). The most common class of HABs blanketing the coast of California, Oregon, extending into Washington is *Pseudo-nitzschia* [3]. The marine diatom is known to produce phycotoxin, Domoic Acid (DA) and has been previously identified to be a major economic and health threat. Although, structurally similar to the conventional neurotransmitters, toxicity of DA predominantly arises from its higher binding affinity to the N-methyl-d-aspartate (NMDA) receptors on the neurons [4]. DA binding leads to increased calcium ion flux, unregulated enzyme activity, swelling of neuronal cells and eventually cell death. Human consumption of DA contaminated marine food sources such as razor clams, mussels, and Dungeness crabs has been associated with a variety of symptoms ranging from gastrointestinal distress to brain damage and in rare cases, even death [5]. Progressive accumulation and consumption of DA-contaminated food thus classifies as a potential public health emergency particularly in view of the frequent algal outbreaks.

Several marine health monitoring programs lay importance in developing strategies for early warning and predictive capacity as preventative or mitigating measure to avert HABs, at least the scale of it. Early detection and predictive sensitization can both be achieved by continuous monitoring or frequent testing of DA contaminated food samples, phyto-planktons and seawater. Conventional screening methodologies, including high performance liquid chromatography in tandem with mass spectrometry (HPLC-MS), bioassays, fluorescence sensors and Enzyme-linked immunosorbent assays (ELISA) are sensitive, but difficult to adapt for the purpose [6,7,8]. Their restrictive features include device complexity, requirement of skilled personal, expensive infrastructure, high cost of operation and elaborative extraction protocols implicating requirement for testing alternatives. Surface-enhanced Raman spectroscopy (SERS) presents a viable solution to address the limitations of traditional sensing methods. SERS offers sensitivity, molecular specificity, cost effectiveness, portability and non-invasiveness [9]. Additional advantage of using SERS based DA screening includes the potential to mark structural and functional changes. For instance, hydration in DA molecules leads to shift in geometry of DA side chain, altering the SERS spectrum [10]. The functional information gathered from the changed spectral features indicate decreased activity of DA on account of loss of affinity for binding, an attribute mostly absent in other detection techniques. SERS based DA screening finds scattered and limited mention in literature, mostly restrictive to presence or absence of DA [11,12,13]. No study, to the best of our knowledge, has been presented to achieve quantitative estimation of DA from crab meat, which forms the focal point of discussion for the presented article.

Through this work, we propose a hybrid testing model, based on SERS and multivariate analysis for quantitative screening of DA from spiked water and real crab meat samples. SERS substrates using diatomaceous earth (DE) enriched by silver nanoparticles (AgNPs) were used to acquire characteristic DA vibration spectrum. The hybrid, AgNPs-diatom or AgNPs-DE have been previously proven to be excellent SERS substrates [14,15,16]. The periodic porous patterns along the diatom skeleton not only allows dense arrangement of nanoparticles on its surface generating ‘hot spots’, but are also known to increase mass transport onto the SERS substrate [17]. The experimental data obtained over testing a range of DA concentrations were utilized to develop a standard calibration curve for DA as a function of SERS signal intensity. The testing results were tabulated over different substrates and multiple data collection points to rule out the probability of data degeneracy owing to any irregularity of SERS substrates. The estimated relative standard deviation for each tested concentration was <20%, falling well within the statistically acceptable range. The developed reference data were later utilized to predict DA concentrations from real crab meat samples, both control and artificially spiked. The elution of spiked DA was realized using simple ultra-sonication and centrifugation steps, reducing process complexity. The limit of detection achieved was similar to previous literature. Analyzed by Principal component analysis (PCA) and Partial least square regression (PLSR) models, statistical correctness and quantitative estimations were validated. Experimental results complimented with statistical modelling proved the effectiveness of the developed cost-effective testing platform to detect and quantify DA in shellfish samples.

## 2. Results and Discussion

### 2.1. Characterization of SERS Substrates

The morphological details of the fabricated AgNPs-DE SERS substrates were studied using scanning electron microscope (SEM). The representative micrographs obtained are presented as Figure 1a–c. The diatomaceous component contributing to the SERS substrate were typically found to be round, resembling a disk, with distinguished presence of pores and cavities running through-out the structure (Figure 1a (inset)). The average pore size was estimated to be <200 nm. The substrates were also marked by abundant presence of spherical nanoparticles distributed in a non-uniform, but regular, manner throughout the substrate. Energy Dispersive Spectroscopy (EDS) and elemental mapping were used to determine the chemical composition and physical localization of constituting components on the substrate. As observed in the EDS spectrum in Figure 1d, strong reflections from characteristic energy values of Si, O and Ag, indicating presence of silica, oxygen and silver, were marked [18]. No additional peaks were observed suggesting sample purity. The elemental distribution maps of the AgNPs-DE substrate (Figure 1e–h) display a unique color code reference, identifying each contributing element. The larger spherical porous structures were seen to reflect signals for silica and oxygen, while smaller functionalizing nanoparticles reflected silver. No other interfering signals were mapped affirming diatom-Ag presence and lack of impurities in the sample.

### 2.2. SERS of DA from Spiked Water

DA is a tribasic amino acid. Assignment of the typical Raman reflections in DA are based on the vibrational information from group wave numbers (CO, NH_2_^+^, COOH) in structurally resembling amino acids including proline, glycine, glutamic acid and alanine. In crystalline form, DA exists as a zwitterionic configuration and consists of four characteristic (NH_2_^+^) modes appearing at 853, 1195, 1387, 1562 cm^−1^ and asymmetric and symmetric stretching vibrations for carboxylate group (νCOO) at 1616 and 1414 cm^−1^, respectively [10].

The Raman spectrum acquired for the aqueous solutions of DA for our study shows some significant changes to the characteristic Raman vibrational map of the crystalline DA (from literature). Figure 2a shows the recorded SERS spectrum of DA aqueous solutions in a series of concentration ranging between 1 and 1000 ppm. The acquired spectrum is identified with two major vibrations centered at 1609 cm^−1^ and 1378 cm^−1^, representing the C=C in the ethylenic group and NH_2_^+^ twisting (τ NH_2_^+^) mode, respectively. The weaker vibrational reflections centered at 937 and 952 cm^−1^ are well indexed to DA but have been degenerately associated with the stretching vibrations in the C-C-N (ν_CCN_) and C-N (ν_CN_) bonds. The shouldering peak at 1266 cm^−1^ is red shifted from conventional peak position at 1278 cm^−1^ and is indexed to the presence of in-plane deformation of OH in COOH. The Raman spectrum of aqueous DA is not only peak shifted compared to solid DA but also appears typically broader than usual. The significant peak shifting and broadening of the Raman spectra is indicative of the interaction between the DA molecules and the bio-photonic crystal-Ag hybrids on the SERS substrates. Interactions between NH_2_^+^ group with neighboring carboxylate group or water molecules through hydrogen bonding might be a contributing factor to the observant shift in peak positions. Similar observations have been marked and reported previously [19,20].

The Raman spectra acquired during the study were later utilized to develop a standard data curve which was used to screen and quantify DA from real crab meat samples. The developed standard curve plotted as Raman intensity versus DA concentration is presented as Figure 2b. To test the robustness and reproducibility of the fabricated substrates, multiple Raman spectra were collected from random loci on the substrates. Each data point on the standard curve, corresponding to a specific DA concentration, represents an average value tabulated over 10 different spectra. For each tested concentration, the estimated standard deviation and average intensity (horizontal bar) for Raman signature at 1609 cm^−1^ was also calculated and is presented in Figure 3. The tabulated RSD values < 20% suggests good reproducibility and statistical significance of the acquired data.

### 2.3. Multivariate Analysis—PCA and PLSR Study

Prior to utilization of the standard reference curve for practical application, the acquired data were also tested using mathematical models, PCA and PLSR. PCA study was focused to testify the standardization of the acquired data set and to simplify the correlation between different spectral data sets by maintaining the usefulness of the data derived from key contributing variables only. The analysis was conducted by extracting principal components from wavenumbers ranging between 1200 and 1800 cm^−1^ for all working DA concentrations from 1 to 1000 ppm. The modelled PCA curve, as presented in Figure 4a, could evidently separate the spectrum variables as a function of DA concentration. The first three Principal components (PCs) could describe 98% of the variance for different DA concentrations, with PC1 showing maximum contribution accounting for majority of score at 93.01%. Each data point for a particular concentration is represented with a sphere of unique color. At higher DA concentrations, when the number of molecules contributing to the data are abundant, low to negligible separation within a data set is observed. However, as the DA concentration decreases and the Raman spectra moves away from ensemble averaging, the contribution from each molecule to the spectrum attains significance and less proximity within the data points of a particular concentrations are observed. This observation holds similarity to the calculated RSD data where the deviation values though within the acceptable limits, show higher variance. To assist accurate prediction of DA concentration for quantitative analysis, root mean square error-based metric was also used for the study. PLSR model with a five-fold cross validation, 80% random selectivity for training data and 20% for test data was performed. The trained model received an estimated RMSE value of 0.99 while for test data set it was 0.89, relating closely to our observations from RSD and PCA models. The representing spectrum is presented as Figure 4b.

### 2.4. SERS from Crab Meat Extract

To examine the utilization potential of the synthesized AgNPs-DE substrate towards DA detection in a real on-site setting, we screened the SERS activity of DA eluted from spiked meat samples. Figure 5a shows the SERS spectrum of DA eluted in meat extract at different spiking concentrations ranging from 1 to 1000 ppm. In a parallel set up, a control sample without any external DA spiking was also maintained and tested for DA presence, if any. General features from the acquired spectra were mostly similar to those obtained for DA spiked water (Figure 2); however, a set of additional peaks centered around 1299, 1340 and 1408 cm^−1^ was also observed. The designated reflections are identified to C-C stretch and amide III groups in proteins and can be contributed through different biological matter eluted in the meat extract [21].

Notably, the intensity of the Raman signal corresponding to a particular concentration was found to be lower than that observed in the developed standard spectrum. The signal drop could possibly arise from (a) simplicity of the testing methodology, as no affinity binding or separating channels/columns were used, (b) accumulation of DA molecules within deep tissues, which are difficult to extract using simple treatment used in our work, and/or (c) signal masking in the presence of additional biological molecules eluted during the extraction process [22,23]. Irrespective of the loss of signal strength, the elution efficiency for each tested concentration ranged between 78 and 90%, following an acceptable deviation standard. For individual concentrations tested, the results are tabulated as Table 1.

## 3. Materials and Methods

### 3.1. Chemicals and Materials

Diatomaceous earth (Celite^®^ 209), Sodium Carboxymethyl cellulose (Na-CMC), Silver nitrate (AgNO_3_) and L-Ascorbic acid (L-AA) were purchased from Sigma Aldrich, St. Louis, MO, USA. Stannous chloride (SnCl_2_) and Hydrochloride acid (HCl) were procured from Alfa Aesar. The biotoxin standard, Domoic Acid was purchased from MuseChem, Arrakis Tek Inc., Fairfield, NJ, USA. Dungeness crab was brought from seafood market in Newport, situated along Oregon’s central coast. All the solutions and dilutions were prepared in ultrapure Milli Q (18.2 MΩ cm) until mentioned otherwise.

### 3.2. Fabrication of SERS Substrates

#### 3.2.1. Diatomaceous Earth Pretreatment and Substrate Coating

Diluted solution of as received DE powder was sieved sequentially through regular Whatman^®^ filter papers of two different pore sizes, 20 and 11 µm, to obtain a filtrate rich in porous diatom skeleton with uniform shape and size. The filtered DE was then dried and activated by heating in a hot air oven for 2 h maintained at 200 °C. The treated DE was stored in an airtight glass vessel and used as required.

For substrate coating, an aqueous DE solution was prepared by mixing 12 g of treated DE with 50 mg of CMC in 20 mL of Milli Q. The solution was thoroughly vortexed until DE was completely mixed; the solution attained uniform consistency and no DE powder lumps were remaining. The traditional cover glasses (22 × 22 mm) were used as the solid substrate to coat the DE-CMC mix using a spin coating system. For each substrate, equal aliquots of DE-CMC were poured onto the cover glasses and spun for 25 s at 50 and 100 rpm, in succession. The prepared substrates were thereafter maintained in the hot air oven at 100 °C for 30 min or until dry. The dried substrates were placed in cleaned petri dishes and sealed using parafilm for long term use. 

#### 3.2.2. Functionalization of DE Substrates—In Situ Growth of Silver Nanoparticles

Functionalization of the DE substrates was achieved using an in situ growth protocol previously optimized and published by our group [24,25,26]. Briefly, the DE coated substrates were soaked in a 1:1 solution of SnCl_2_ and HCl (20 mM) for 30 min to allow effective binding of Sn^2+^ ions on the silica rich diatom skeleton. The Sn^2+^ binding sites serve as nucleation centers for silver nanoparticle seeding. After SnCl_2_-HCl soaking, the substrates were washed in Milli Q followed by baking in the hot air oven. Each step was maintained for 10 min. The dried substrates were then immersed in an aqueous solution of 20 mM AgNO_3_ for 30 min to assist ion replacement and allow Ag^+^ deposition and seeding. The substrates were removed from the silver solution and washed in Milli Q again for 10 min to remove any excess/unbound ions on the substrate surface. Following drying, the substrates were soaked in the growth solution for the next 30 min to allow nanoparticle growth on the seeded substrate. The composition of the growth solution used was L-AA (50 mM):AgNO_3_ (5 mM) *v/v* at 1:2. Silver nanoparticle functionalized DE substrates were washed in Milli Q, dried and stored under vacuum conditions to counter any damage due to surface oxidation.

### 3.3. Raman Measurements

#### 3.3.1. Instrumentation

All the SERS measurements were performed on the BWTek iRaman Plus^®^ (BWS465-5328, Metrohm, FL, USA) portable Raman system, equipped with a 532 nm laser source. For each measurement, the laser was focused at a working distance of 5 mm and the intensity was maintained at 25% of the maximum power (50 mW). Integration time and number of accumulations per measurement was 60 s and 2, respectively. For each concentration of DA tested, to statistically validate the results, Raman spectra were acquired from 3 SERS substrates fabricated in a single batch. On each substrate, a minimum of 10 measurements were performed. In all, each data point in the calibration curve is an averaged value over 30 Raman spectra for the corresponding DA concentration.

#### 3.3.2. Domoic Acid Detection in Spiked Water

The as received DA formulation was diluted to prepare a stock solution with an effective concentration of 1000 ppm. The stock was serially diluted to prepare a set of DA working solutions in concentration ranging between 1 and 1000 ppm. For every testing concentration, the SERS substrates were soaked in 5 mL solution and allowed to attain saturation overnight. Following soaking, the substrate was removed from the dipping solution and dried in a desiccator prior to measurement.

#### 3.3.3. Domoic Acid Detection in Crab Meat Extract

Crab meat bought from the local market was cleaned and rinsed thoroughly with water. The washed meat was ground in a blender to obtain a near homogenized starting mix. A total of 7 portions of homogenized meat samples, 10 g each, were weighed and added to falcon tubes. Each falcon tube contained 10 mL of DA working solution of different concentrations being tested (1–1000 ppm). The DA samples were left to be soaked up in the meat tissue overnight. To avoid any degradation of DA, the overnight set up was maintained at room temperatures (15–20 °C) and under dark conditions. The overnight soaked meat samples were filtered using Whatman^®^ filter paper to separate any unabsorbed DA, which was negligible as the meat appeared slimy and swollen, indicating imbibition of aqueous solution. The filtrate containing artificially spiked DA enriched meat samples were placed in fresh falcon tubes with 5 mL of Milli Q, used here as the elution medium to minimize any interference to DA signal. This step was repeated for all 7 concentrations of DA being tested. To elute DA, the falcon tubes were subjected to a cycle of ultra-sonication (60 °C, 15 min) followed by centrifugation (3000 rpm, 10 min). The supernatant was collected for analysis. Prior to using the eluted samples for soaking the SERS substrates, the eluted supernatants were filtered through 0.2 µm PTFE (VMR) filters. DA working solutions and substrate soaking were performed similarly as described for spiked water samples. In addition to the 7 DA concentrations eluted from the spiked meat samples, a control sample was also maintained, where the meat sample was not soaked in DA but Milli Q. All the experimental steps were performed identically to DA soaked samples to maintain the accuracy of the experimental conditions.

### 3.4. Spectral Analysis

To complement our experimental findings, multivariate statistical analysis was also performed. The statistical studies were performed using the data plotting software Origin Pro 9 and MATLAB. Both Principal component analysis (PCA) and Partial least square regression (PLSR) accentuates the participation of core contributing factors in a large data set while limiting any data degeneracy from participating variables with smaller or negligible contribution.

### 3.5. Characterization

AgNPs-DE SERS substrates were visualized on a scanning electron microscope FEI 196 Quanta 600 FEG system. The electron micrographs were obtained at an operational accelerating voltage of 15–30 kV.

## 4. Conclusions

Portable Raman spectroscopy was utilized as a facile route to quantitatively sense marine toxin, Domoic Acid, independent of tedious and expensive extraction protocols. The limit of detection achieved through the work finds proximity to the current literature. The SERS spectra of different concentrations of DA aqueous solutions were well identified but greatly varied in nature with degree of protonation and hydration, suggesting strong interaction of the analyte molecules with the silver nanoparticle enriched bio-photonic substrates. Following the regression modelling of the calibration curve, obtained through the experimental study, concentration of DA eluted from real crab meat samples were predicted. Multivariate analysis, Principal Component and Partial Least Square Regression were modelled to statistically validate the correctness of the obtained data values from control as well as the test samples. The significance of the study is drawn from its easy-to-use approach, speed, sensitivity, portability and the potential of being developed into an economical on-site seafood monitoring tool.

## Figures and Tables

**Figure 1 molecules-27-08364-f001:**
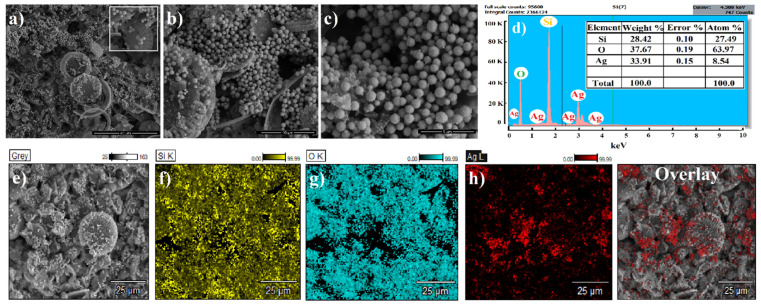
(**a**–**c**) SEM images of the fabricated AgNPs-DE SERS substrates acquired at different magnifications, (**d**) representative EDS spectrum and (**e**–**h**) Elemental mapping of the constituting components of the substrates. Si, O and Ag are the only contributing signals, while Si and O are scattered throughout the plane of view, Ag appears to be localized in small pockets. The unique color code yellow, cyan and red represent Si, O and Ag, respectively.

**Figure 2 molecules-27-08364-f002:**
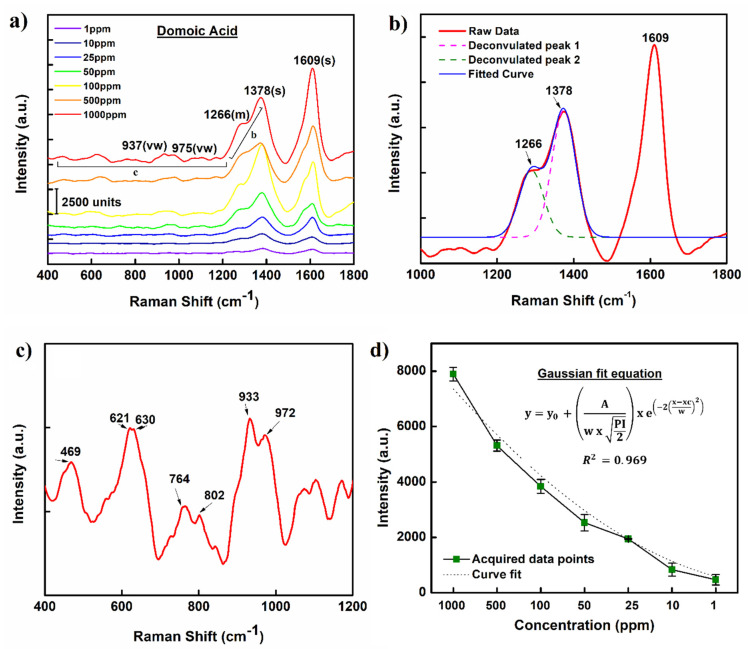
(**a**) Raman spectrum of DA at different concentrations (1000−1 ppm) acquired on AgNPs−DE SERS substrates. (**b**−**c**) Enlarged view of the acquired DA Raman spectrum over selective wavenumbers for detailed peak analysis. (**d**) Calibration curve representing relationship between DA concentration and Raman signal intensity. Notations s, m and vw represent strong, medium and very weak signal intensity respectively.

**Figure 3 molecules-27-08364-f003:**
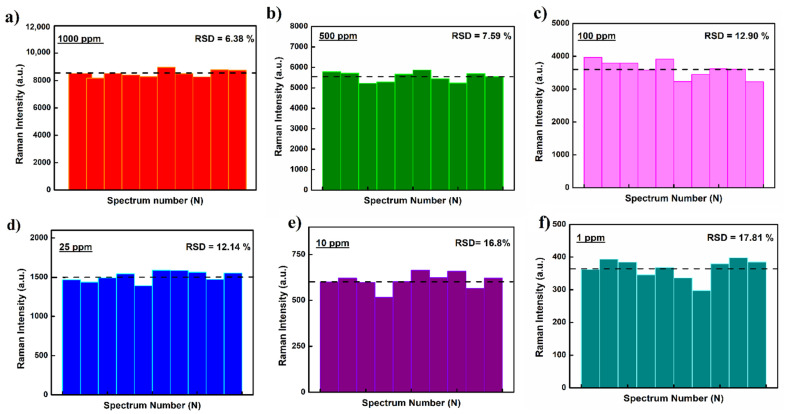
(**a**−**f**) SERS intensity variability for tested DA concentrations collected at random locii on the fabricated AgNps-DE substrates. RSD = relative standard deviation of the SERS signal. The dotted black line represents the average intensity value for peak reflection at 1609 cm^−1^ for respective tested concentration.

**Figure 4 molecules-27-08364-f004:**
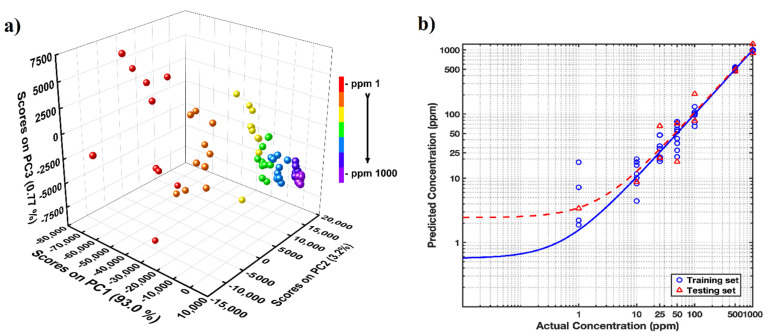
(**a**) PCA Scatter plot for different concentrations of DA; (**b**) PCA−PLSR calibration curves used for DA quantification from spiked DA water.

**Figure 5 molecules-27-08364-f005:**
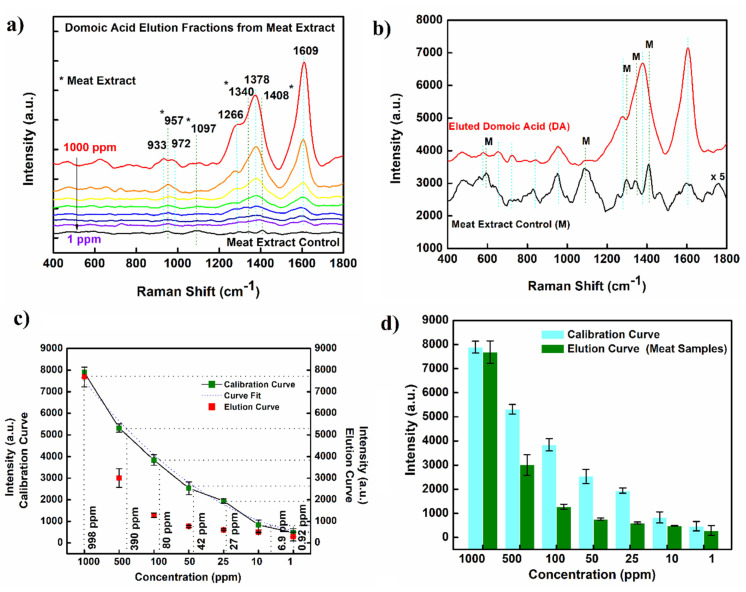
(**a**) Raman spectrum of different concentrations, 1000 ppm (red) to 1 ppm (violet) of DA eluted from spiked crab meal samples; (**b**) Enlarged view of the Raman spectrum acquired for eluted DA and control meat sample comparing overlapping signals; (**c**) Estimation of DA concentration from eluted samples; (**d**) Histogram of Raman signal intensity as acquired from standard data curve versus eluted samples. Notations asterisk (*) and M represent vibrational reflections from meat samples.

**Table 1 molecules-27-08364-t001:** DA concentrations tabulated for the eluted samples from the SERS calibration curve.

S. No	Spiking Concentration (ppm)	Concentration (ppm) Tabulated from SERS	Tabulated Elution Efficiency (%)
1.	1000	998	~78–90%
2.	500	390	
3.	100	80	
4.	50	42	
5.	25	27	
6.	10	6.9	
7.	1	0.92	
8.	Control	<5 ppm	

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
