# Peer review of "Quantitative Sensing of Domoic Acid from Shellfish Using Biological Photonic Crystal Enhanced SERS Substrates"

_molecules, 2022, doi:10.3390/molecules27238364_

Round 1

Reviewer 1 Report

In this manuscript, the authors addressed the limit of detection achieved was similar to previous literature. Analyzed by Principal component analysis (PCA) and Partial least square regression (PLSR) models, statistical correctness, and quantitative estimations were validated. Experimental results complemented with statistical modeling proved the effectiveness of the developed cost-effective testing platform to detect and quantify DA in shellfish samples.

There are minor suggestions as follows: 

1. In the abstract, there is a need to reduce the length of the abstract and address the main results of the study.

2. For the confirmation of extracted polymer whether authors used some spectroscopic analysis such as NMR etc.

3. Please separate statistical section in materials and methods.

4. References need to become in the format of the journal. 

Author Response

The authors are thankful to the reviewers for their suggestions regarding the submitted manuscript. The requisite changes to the main article and the answers to the posed questions are shared herewith. We certainly believe, the recommended changes will help achieve an article with improved clarity and relevance. We look forward to the acceptance of the article.

Comments and Suggestions for Authors

Reviewer 01

In this manuscript, the authors addressed the limit of detection achieved was similar to previous literature. Analyzed by Principal component analysis (PCA) and Partial least square regression (PLSR) models, statistical correctness, and quantitative estimations were validated. Experimental results complemented with statistical modeling proved the effectiveness of the developed cost-effective testing platform to detect and quantify DA in shellfish samples.

There are minor suggestions as follows: 

  1. In the abstract, there is a need to reduce the length of the abstract and address the main results of the study.

Response: Thank you for the suggestion, the revised abstract highlighting the main results from the study is now included in the main manuscript.

  1. For the confirmation of extracted polymer whether authors used some spectroscopic analysis such as NMR etc.

Response: We thank the reviewer for presenting with the very useful suggestion of using spectroscopy to validate our results and findings. Unfortunately, we could not perform the said experiments within the stipulated time for submission of the revised article. As sophisticated instruments such as NMR, HPLC are maintained as a central facility in our university, it was difficult to earn a characterization time. We would like to request the reviewer to kindly excuse us on this suggestion; we have made a note of it and will include it in our next set of experiments for the study.

  1. Please separate statistical section in materials and methods.

Response: The details of statistical experiment has been included as a separate section titled  “2.4 Spectral Analysis”in the revised manuscript.

  1. References need to become in the format of the journal. 

Response: Thank you for bringing it to our notice, the requisite changes have been made. The revised article includes the references in the journal format.

Reviewer 2 Report

In this short communication, authors presented they prepared AgNps-DE for SERS detection of DA in spiked crab meal samples. It is a concerned issue for seafood safety and people health. However, the experimental results lack convincing.

1.Average size of AgNps should be estimated and SPR result should be provided.

2.What is the shelf-time of AgNps-DE? It is a quite imperative performance for SERS real application.

3. The description on the detail to obtain the elution efficiency is missing.

4. What is the limitation on DA residue level for seafood?

5. The caption of Figure 2 is unclear and should be revised to supply complete information for each graph.

6. Normal Raman spectrum of crystal DA must be offered for reference.

7. In Figure 2, the significant peak shifting and broadening of the Raman spectra of DA on AgNps-De are seemly due to carbonization of samples. Therefore, laser-heating effect must be concerned and avoided.

8.HPLC method should simultaneously be conducted for validating the robustness of SERS assay.

9. The unit for resistance of ultrapure water is MΩ cm.

10.Put a space between number and unit.

Author Response

The authors are thankful to the reviewers for their suggestions regarding the submitted manuscript. The requisite changes to the main article and the answers to the posed questions are shared herewith. We certainly believe, the recommended changes will help achieve an article with improved clarity and relevance. We look forward to the acceptance of the article.

Reviewer 02

In this short communication, authors presented they prepared AgNps-DE for SERS detection of DA in spiked crab meal samples. It is a concerned issue for seafood safety and people health. However, the experimental results lack convincing.

1.Average size of AgNps should be estimated and SPR result should be provided.

Response: We thank the reviewer for the suggestion. The average particle size of the functionalizing AgNps was estimated using the SEM images and the software, Image J. The tabulated size was 800±76.34 nm. As suggested, UV-Visible spectroscopy was also performed. The representative spectrum is presented as Figure S1. The absorption character was observed to be visible light responsive with a peak maximum at 426 nm, indexing well to the characteristic LSPR of the silver nanoparticles. An additional absorbance peak, with small intensity was marked at longer wavelength ~600 nm, ascribing to the electromagnetic feature in the quasi-aggregated cluster morphology of the functionalizing particle (inset) [R1].

Figure S1. UV-Visible spectrum and high magnification SEM images of the functionalizing AgNps

2.What is the shelf-time of AgNps-DE? It is a quite imperative performance for SERS real application.

Response: We thank the reviewer for the question. We agree to their comment about substrate stability being imperative to performance. All the substrates were fabricated in a single batch and maintained at room temperature. The tests were performed over a period of 30 days. No significant errors or shift of spectrum owning to substrate degradation was found within this period. To minimize loss of any activity on account of surface oxidation, prior and post usage, the substrates were stored in a glass petri dishes sealed with parafilm.

  1. The description on the detail to obtain the elution efficiency is missing.

Response: The authors are thankful for the question. The elution efficiency was based upon the intensity calibration curve obtained by the performed SERS experiments. Using the mathematical expression (1), concentration of DA retained within the tissue or lost during the cleaning/extraction process was estimated. The %loss was then used to find the effective elution using unitary method.

     (1)

  1. What is the limitation on DA residue level for seafood?

Response: The regulatory limit of DA in crab meat is <20 ppm while in terms of its potency to cause Amnesic Shellfish Poisoning (ASP); the limit if drawn at 20µg DA/gram tissue [R2].

  1. The caption of Figure 2 is unclear and should be revised to supply complete information for each graph.

Response: We apologise for presenting incomplete information and causing confusion. A revised caption with detailed description has been included in the manuscript.

  1. Normal Raman spectrum of crystal DA must be offered for reference.

Response: We thank the reviewer for the suggestion, however, being a controlled substance, DA is manufactured and supplied as small aliquots of aqueous solution only, obtaining a normal Raman spectrum of crystal DA is difficult.

  1. In Figure 2, the significant peak shifting and broadening of the Raman spectra of DA on AgNps-De are seemly due to carbonization of samples. Therefore, laser-heating effect must be concerned and avoided.

Response: The authors are thankful to the reviewer for such a meaningful suggestion. A series of experiment aiming to optimize measurement parameters for efficient DA screening was performed (data not included in the main manuscript). A representative spectrum measuring change of DA SERS characteristics as a function of laser power is presented as Figure S2. As observed from the acquired spectrum ranging from lowest to highest power, the basic character of the DA Raman spectrum was consistent except for signal intensity that varies linearly with increasing laser power. We agree with the reviewer that sample carbonization could result in peak broadening however based on our experimental data, we fail to recognise any considerable shift in DA SERS spectrum due to laser heating.

Figure S2: SERS spectrum of DA on synthesized substrates obtained over a range of different laser power intensity to study the effect of laser heating.

8.HPLC method should simultaneously be conducted for validating the robustness of SERS assay.

Response: We are grateful to the reviewer for their valuable suggestion on performing the HPLC experiment to test the robustness of the SERS assay. We did attempt to study the HPLC results for our tests however lost the sample due to DA degradation. A second characterization date could not be obtained with the central facility with the limited time we had prior to submission of the revised article. We would like request the reviewer to kindly excuse us for our inability to re-run the experiment due to limited resources and time at hand. We have made a note of the suggestion and will include the HPLC data in our following studies under the continuing DA project.

  1. The unit for resistance of ultrapure water is MΩ cm.

Response: We apologize for the mistake; the correct unit has been included in the revised manuscript.

10.Put a space between number and unit.

Response: Thank you for bringing it to our notice, the requisite changes have been made.

References

R1. Jiang, X.; Zeng, Q.; Yu, A. A self-seeding coreduction method for shape control of silver nanoplates. Nanotechnology 2006, 17104, 4929-4935.

R2. Wekell, J. C.; Hurst, J.; Lefebvre, Kathi. The origin of the regulatory limits for PSP and ASP toxins in shellfish. Journal of Shellfish Research 2004, 23, 927-930.

Round 2

Reviewer 2 Report

The revision is improved and I understand the difficulty of the experimental condition. I recommend to accept the revision for publication as it is.